# Study of Natural Longlife Juvenility and Tissue Regeneration in Caudate Amphibians and Potential Application of Resulting Data in Biomedicine

**DOI:** 10.3390/jdb9010002

**Published:** 2021-01-18

**Authors:** Eleonora N. Grigoryan

**Affiliations:** Kol’tsov Institute of Developmental Biology, Russian Academy of Sciences, 119334 Moscow, Russia; leonore@mail.ru; Tel.: +7-(499)-1350052

**Keywords:** salamanders, juvenile state, tissue regeneration, extracts, microvesicles, cell rejuvenation

## Abstract

The review considers the molecular, cellular, organismal, and ontogenetic properties of Urodela that exhibit the highest regenerative abilities among tetrapods. The genome specifics and the expression of genes associated with cell plasticity are analyzed. The simplification of tissue structure is shown using the examples of the sensory retina and brain in mature Urodela. Cells of these and some other tissues are ready to initiate proliferation and manifest the plasticity of their phenotype as well as the correct integration into the pre-existing or de novo forming tissue structure. Without excluding other factors that determine regeneration, the pedomorphosis and juvenile properties, identified on different levels of Urodele amphibians, are assumed to be the main explanation for their high regenerative abilities. These properties, being fundamental for tissue regeneration, have been lost by amniotes. Experiments aimed at mammalian cell rejuvenation currently use various approaches. They include, in particular, methods that use secretomes from regenerating tissues of caudate amphibians and fish for inducing regenerative responses of cells. Such an approach, along with those developed on the basis of knowledge about the molecular and genetic nature and age dependence of regeneration, may become one more step in the development of regenerative medicine

## 1. Introduction

A key issue in biology and medicine is the molecular/genetic and cellular bases of the animals’ and humans’ ability to regenerate organs and tissues. The major conclusion drawn from many years of research is that the regenerative capacity of animals decreases as they age and in the evolutionary series. Despite the many exceptions and interspecific differences in the manifestation of the regenerative capabilities, as well as the use of different mechanisms to regenerate lost tissues, this rule remains fundamental—the simpler and younger the organism, the more successfully it restores lost or damaged parts. Furthermore, there are tetrapods which being mature are able to regenerate tissues and this ability is not lost with age. These are mature caudate amphibians (Urodela or Caudata). Such animals provide researchers with the organ and tissue regeneration models that help understand the cellular and molecular bases of the phenomenon. The regenerative abilities, evolutionarily fixed in the Urodela, have various explanations. Although a major part of research has been aimed at understanding the cellular and molecular mechanisms of regeneration in these animals, very little research has been addressed to the issue of how such fundamental organismal traits as genome size, immunity, low metabolic rate and rate of cell differentiation, or stage of the life-cycle may affect regenerative potential. Figure 1 shows the main features of Urodela associated with their high regenerative capacity.

Below we consider the features of the developmental status of caudate amphibians, manifested in the properties of genome operation, in tissue structures, and in cell behavior, that contribute to regeneration. In addition to comprehensive studies on the external and internal regulation of cellular and molecular mechanisms of regeneration in the Urodela, the examples of how regenerating tissues are used as a source of factors that can stimulate regenerative processes in tissues of higher vertebrates are also discussed.

## 2. Immunity, Genetic Features, and Cell Plasticity Associated with Regenerative Abilities in Urodela Amphibians

Caudate amphibians have the capability of not only tissue, but also organ regeneration that is the most pronounced among vertebrates [1,2]. Newts and axolotls can restore limbs, tails, fragments of the spinal cord and brain, jaws, intestines, the heart ventricles, and also some eye tissues: the lens, the retina, and the optic nerve. Adult higher vertebrates and humans lack these abilities. For this reason, attempts to explain them have been made since the discovery of the phenomenon.

It is currently known that the ancient and highly conservative immune system can not only regulate, but also be directly involved in tissue regeneration in animals, including caudate amphibians [3,4]. Its relationship to the processes of regeneration of various tissues and organs in salamanders is considered in detail in a recent review [4]. According to the summarized information, the innate and adaptive components of the immune system in the Urodela are found at critical stages of regeneration of many tissues, from the moment of damage to the morphogenesis of newly formed regenerates. The role of macrophages, the direct participants in the regeneration of the heart and limb of these animals, is also reported. Particular attention is paid to the complement system and its plasma proteins, which also play a significant role in the regeneration processes [4].

The specifics of immune response in caudate amphibians and the long healing without visible inflammation and scarring were previously reported by Mescher and Neff [5,6]. The authors considered this issue in the context of the evolution of immature and adaptive immune systems. Crucial differences in these characteristics were noted within the class Amphibia. The Urodela, compared to the Anura (ecaudate amphibians), not only manifest a simpler organization, but also have a certain “immunodeficiency”: a slow immune response, weak immune memory, and cellular immunity [7]. In the Anura, the regenerative ability correlates with the course of ontogenetic development of the adaptive immune system [8]. The regeneration success is determined by a set of interactions between the cells that are regeneration sources on the one hand and the cells and factors of the immune system on the other [8]. An inverse relationship is assumed between the maturity of the immune system and the capability of regeneration [9].

It is also important that, in contrast to mammals, the Urodela possess a highly effective mechanism of cell senescence immunosurveillance [10]. Aging cells (including stem cells) are accumulated with age, which hampers regeneration even where it is fundamentally possible [10]. Thus, skeletal muscle satellite cells responsible for the muscle fiber regeneration are depleted in mammals as they age [11,12]. On the other hand, the accumulation of aging cells in higher vertebrates, which occurs due to DNA damage with age, the accumulation of free radicals, the telomere shortening, etc. induces aging of neighboring cells. Caudate amphibians, having the mechanism of senescence immunosurveillance, prevent not only their accumulation, but also de novo emergence. In the case of limb regeneration in salamanders, the recurrent emergence of aging cells occurs along with the formation and expansion of progenitor cells, while aging cells are quickly eliminated by macrophages [13]. In contrast, the role of macrophages that inhibit regeneration in ecaudate amphibians is known to be performed at the developmental stages prior to metamorphosis, the time when the limb regeneration in these animals, unlike that in salamanders, is no longer possible [14]. It cannot be ruled out that the immunosurveillance that occurs in amphibians during regeneration is taxon-specific—it stimulates regeneration in the case of Urodela and inhibits in the Anura.

Other explanations for the high regenerative capacity of the Urodela are based on the study of differentiation of cells that are sources of regeneration, or rather its plasticity, found in many cell types of these animals. Plasticity is defined as the possibility to lose the traits of specialization (dedifferentiation) by the initial, differentiated cells, the entry into the proliferative phase, and the formation of populations of amplifying cells similar to stem-like ones that can acquire new differentiation and restore the lost tissue when they exit the replication cycle [15]. The classical examples are the conversion of pigmented epithelial cells, the iris and retinal pigment epithelium (RPE) of newt eye into, respectively, lens and retinal cells. This transformation, referred to as “transdifferentiation” [16], occurs through the formation of transitory populations of actively dividing dedifferentiated cells. This feature (a high degree of cellular plasticity) is lost with development in birds and mammals [15]. An alternative to cell plasticity is the terminal differentiation that stabilizes with age. The process is based, in particular, on changes in the cell epigenetic landscape [17,18].

Various epigenetic mechanisms, such as DNA methylation, histone modifications, non-coding RNA activity, etc., are known to be involved in both the formation and change of cell phenotype through increasing, decreasing, or silencing gene activities. Data on the epigenetic status and its variations in the cells that are regeneration sources in the Urodela, which can shed light on the molecular nature of both dedifferentiation (phenotypic rejuvenation) and stabilization of the original or newly acquired differentiation, are still extremely scarce. Some information has been obtained using the model of eye lens regeneration from iris cells in newts by Maki and coauthors [19,20,21]. They found that histone B4, an oocyte-type linker histone, is required for transdifferentiation of pigmented iris cells into lens fibers. Knocking down of B4 significantly reduces the proliferation of source cells, which causes a considerably smaller lens to form. The B4 knockdown altered the gene expression of key genes of lens differentiation and nearly abolished the expression of lens proteins, γ-crystallins. A study of global histone modification on the same model has shown that histone modifications related to gene activation are differentially regulated in the process of iris cell dedifferentiation [21]. There is evidence indicating the necessity of demethylases to be involved in the caudal fin and heart regeneration in zebrafish [22,23].

Besides the explanations for regeneration in Urodela above, its success in these animals is also suggested to have been determined in the course of the microevolutionary pathway. This opinion [24] is based on the discovery of Prod1 proteins, members of the three-finger protein (TFP) family, which are typically found in salamanders and axolotls and are a key regulator of growth and morphogenesis of regenerating limbs. In the range of genes and proteins associated with regeneration that emerged during the microevolution, there are also genes whose expression correlates with limb and tail regeneration in larvae of the African clawed frog *Xenopus laevis* and the caudal fin regeneration in the zebrafish *Danio rerio* [25,26]. The features identified on the genetic level and associated with regeneration in the Urodela are reported in the works of Kumar and co-authors [27,28], demonstrating the key role of the *nAG* (*Agr2*) gene in the process of limb regeneration. In newts, the nAG protein interacts with the Prod1 membrane receptor, which eventually leads to activation of the ERK1/2 MAP kinase signaling pathway in blastemal cells of the regenerating limb [29]. There is also the opinion that in reptiles, birds, and mammals, the genome that had existed in developing larvae and during metamorphosis was later eliminated from the life cycle. The loss of genes utilized for metamorphosis determined also the loss of the capability of organ regeneration in adults [9,30]. However, not all authors share the viewpoint that successful regeneration is directly related to the presence of taxon-specific “regeneration genes” in lower vertebrates. In the study of Suzuki and Ochi [31], they argue that regenerative capacity, on the contrary, cannot be explained by the presence or absence of specific, regeneration-determining genes in the genome. Alternatively, the gene regulatory mechanisms, including changes in epigenetic landscape, may better reveal the molecular basis of regeneration.

In the model of epimorphic limb regeneration in Urodela, a set of basic characteristics of these animals were found to determinate regeneration: body sizes and aging and growth mode [32,33]. It is emphasized that systemic factors regulating the life processes of the whole organism, factors of the blood and immune system, and, to a greater extent, hormones, should be taken into account. Varying in range and concentrations during development and with age, they determine the conditions for cells to be involved in regeneration [33]. In the works [32,33], authors also pay attention to the property of some lower vertebrates to grow in a sexually mature state and respond to “developmental” hormones that circulate in the blood “late” in ontogeny.

## 3. Pedomorphosis, Genome, and Cell Sizes in Urodela

The above explanations for the high regenerative capacity in Urodela are based on phylo- and ontogenetic factors, as it is the case with other animals. A longer larval period, induced by pharmacologically manipulated hormone levels, extended the known timing of use of the regenerative potential in Anura (ecaudate amphibians) [34]. For newborn mice, thyroid hormone was found to play a certain role in stimulating the cardiomyocyte proliferation and increased the probability of prolonged competence of these cells to proliferate during perinatal development [35].

Pedomorphosis in caudate amphibians results from heterochrony. The latter is manifested in two ways—slowing down somatic development with a normal maturation rate (neoteny) or as accelerated sexual development, by completion of which the somatic development also ceases being incomplete (progenesis). In Urodela, pedomorphosis has been identified in both forms (progenesis and neoteny), and both are regulated by thyroid hormone [36].

There are theories explaining the underdevelopment and simplification of certain organs and tissues in salamanders (which, in turn, facilitate regeneration) not only by pedomorphosis, but also by the genome size. Salamanders (as well as lungfishes) are known to have the largest genomes [37]. When compared with the genomes of ecaudate amphibians, the difference can reach tens of times [38,39]. It is now known that such sizes are due to the large number of sequence repeats in DNA and, primarily, because genes have incredibly long introns [40,41,42,43].

It is worth noting that salamander genomes, being largest among vertebrates, show an extremely low level of genetic differences and modifications in the nucleotide structure [39]. A special study of the lymphoid *rag1* gene in several salamander species revealed a very limited number of mutations and substitutions in its nucleotide structure [44]. Lack of mutations is known to reduce the risk of carcinogenesis, and possibly, as an alternative, allows regeneration. It is assumed that lower vertebrates with their highest regenerative potential (fish and amphibians) and animals exhibiting the lowest potential (mammals) have significant differences in their tumor suppressor machinery, which is also associated with differences in regeneration capacity [45,46]. It should also be noted that Urodela amphibians are extremely resistant to spontaneous or chemically induced tumors [47].

The large genome size and the high DNA content may explain a number of features found in Urodela on the cellular level. This is, first, the increased size of cells—they are much larger compared to those of other vertebrates, frequently with high ploidy [48]. Furthermore, a low metabolism rate and also reduced cell division frequency and cell differentiation rate during development have long been known [49]. These properties affect rates of these processes, but not their completeness, which provides not only the regeneration of tissues and organs, but also their functioning.

## 4. Specifics of Eye and Brain Tissues and Their Regeneration in Urodela

The subject of our long-term research has been the Urodela’s unique ability to regenerate the retina damaged in different ways (detachment, cutting of the optic nerve and blood vessels) and even after surgical removal [50,51,52,53,54], as shown in Figure 2. What are the properties of the cell sources for retinal regeneration associated with the unique regeneration ability in salamanders? The retinal pigment epithelium (RPE), the main source of regenerating retinal cells in mature newts, was found to exhibit high plasticity manifested after the retinal damage as the loss of original features, proliferation, reprogramming, and differentiation in neural direction. We associate these processes with the presence of a combination of molecular and genetic properties, characteristic of both specialized RPE cells and their embryonic progenitors (see below), in intact RPE cells [55].

Studies on the salamander neural retina have shown its relatively simple organization [56,57,58]. The “sign” of pedomorphosis in the structure of the newt retina is the presence of “under differentiated”, displaced bipolars with Landolt’s club [58,59]. Furthermore, regions of steady slow growth were found in the retina of newts and mole salamanders [60,61]. In newts, cells on the extreme periphery of the retina can divide and increase their population throughout the lifetime [60]. Cells in this eye region are not morphologically differentiated and express many genes and proteins that are markers of the eye field in the early eye development (*Pax6, Prox1, Six3)* [54]. Expression of the early retinal differentiation regulator, a homologue of *N-Notch*, which is a cell receptor involved in the specification of retinal cells during development, has been found in this region [62].

In regard to the nervous system in salamanders, it has been reported that its morphological differentiation and genome size are inversely correlated [42,63]. The presumable causes are the effect of a low metabolic rate on its development, the “slowing down” of the developmental gene expression, and the epigenetic factors not identified to date [42,64]. The general assumption about the causal relationship between the genome size and the secondary tissue simplification is also confirmed by the fact that the simplicity of the CNS and sensory organs structure has been reported for caudate amphibians and lungfishes, i.e., for animals with the largest genome sizes [65].

The brain of newts and mole salamanders, studied for a long time, is characterized by relative simplicity of its organization [42,66]. According to results of neuroanatomical studies, Urodela lack some parts of the brain characteristic of other tetrapods or they are represented as clusters of neuronal cell bodies [67]. A comparison of ecaudate and caudate amphibians has shown that the former have a much higher number of neurons in *tectum opticum* than the latter, and they are more differentiated morphologically [42,64]. Labeling of proliferating cells with BrdU, a DNA synthesis precursor, revealed that the proliferation rate in the visual region of the brain is also significantly lower in Urodela than in Anura [68]. A less pronounced brain differentiation was observed in Urodela even when compared with cartilaginous and bony fishes [65,66,67,69]. The phenomenon is considered by evolutionists as an example of the secondary simplification of the brain [42,65,66,67].

Studies of salamander brain carried out on a level of gene expression point to its juvenile traits. Expression of Pax6, Pax7, and Pax3 transcription factors (TFs) characteristic of the early eye and CNS development was studied in various parts of the brain and in the retina of adult newts, *Pleurodeles waltl* [69,70]. According to data on the distribution and timing of *pax* gene expression, certain cell populations exhibit embryonic properties by this trait. In the native newt retina, TF Pax6 was detected immunohistochemically in the inner layers of the retina [70]. We have detected the expression of the *Pax6* gene, along with other TFs and signaling molecules, in the retina as links in the regulatory network of the “eye field” during retinal development [55]. These factors were also identified during retinal regeneration from RPE [52,71,72,73]. The co-expression of key molecules of the melanogenic (Otx2) and pro-neural (Pax6, Pitx1(2)) differentiations, as well as signaling regulators (Fgf2 и FgfR2), indicates the probability of “combined” expression of the marker molecules of specific RPE differentiation and the factors regulating the early eye development [54,55,73]. The presence of nucleostemin, the progenitor/stem cell marker protein, in the newt retina is also the evidence that the retinal cells have retained their juvenile properties [74,75].

Lens regeneration from the dorsal part of the iris is classic demonstration of cell transdifferentiation in the newt [16], as shown in Figure 3. The molecular and genetic characteristics of newt iris cells underlie the phenomenon of repeated lens regeneration [76]. Expression of mRNAs encoding key lens structural proteins or TFs (alfa, beta, gamma-crystallins, sox1,2, delta-1, pax6, prox1, and MAFB) after a multi (18 times) lens extraction appeared similar to that of those operated for the first time. Thus, contrary to the belief that regeneration becomes less efficient with time or repetition, repeated regeneration, even at an old age, does not alter newt regenerative capacity [76,77,78]. These unique regenerative abilities are considered as more evidence for the juvenility in Urodela.

The state of juvenility during regeneration in Urodela may probably be maintained in different ways [79]. In the case of lens regeneration, this occurs through retention of the juvenile status of source cells (dorsal region of the iris) with no signs of aging [76,77,78]; in the case of limb regeneration, through replacement of aging cells by new progenitor cells [10]. It seems likely that these different, but commonly reflecting the state of juvenility, mechanisms are involved in the regeneration programs of Urodela. However, in any case, the ultimate cause seems to be the large size of the genome leading to a large cell size and subsequent low rate of cell division.

Pedomorphosis, as a result of heterochrony, can influence the entire organism (the so-called global heterochrony) [80]. In this case, signs of secondary simplification are found in many tissues. The spinal cord in Urodela also shows simplification traits when compared to that in Anura or higher vertebrates. On the morphological level, the main features found are the lack of dorsal and ventral gray horns of the spinal cord and a low level of cell differentiation and migration maintained during the spinal cord formation. Gliocytes in Urodela spinal cord contain simultaneously two types of intermediate cytoskeletal filaments (vimentin and GFAP) [81]. This trait indicates their similarity to the embryonic astroglia of higher vertebrates; in particular, newborn mice and 7-week-old human fetal cerebrum. Thus, the morphological “immaturity” of astroglia in Urodela is another example of juvenile traits retained by a sexually mature organism. Urodela spinal cord contains cells expressing TFs Pax6 and Pax7, the key molecules regulating the development of vertebrate ectodermal structures. These data proved to be similar to the results of a study of the expression of Pax TFs in neotenic mole salamanders [82,83].

Salamanders exhibit the ability to compensate for cell losses in various regions of the brain and spinal cord, even in the mature state [84]. The brain regeneration in Urodela becomes possible due to the neural stem cells, the so-called ependymoglial cells, present in the ventricular regions [85,86,87,88]. Thanks to these cells, salamanders can replace large populations of neurons and restore damaged nerve fibers and the nervous tissue architecture, which eventually allows its structural and functional regeneration. The activation of resident stem cells involves signaling molecules (Hh, BMP, RA, FGF, and Notch) and some neurotransmitters whose expression is characteristic during development [89]. The models for studying the brain regeneration phenomenon in Urodela are described in the literature. Previously, we announced the possibility of active cell proliferation in the visual region of the brain after damage and during retinal regeneration in newts, which was assessed based on the inclusion of a labeled DNA synthesis precursor (^3^H- thymidine) [90]. A series of experiments were carried out with unilateral removal of optic tectum in salamanders [91,92] and partial removal of the dorsal telencephalon in axolotl [93]. In both cases, the ependymoglial cells were able to proliferate, migrate, and differentiate into neurons. In newts, the regeneration not only caused the amount of the optic tectum neural tissue to restore within 6 weeks, but also its laminar structure to form [91,92]. These works are summarized in detail in the review of [94]. A study of the retina regeneration after extensive damage caused by the optic nerve cutting in an adult newt, *Pleurodeles waltl*, has revealed an additional bundle of nerve fibers (nerve) formed in the central part of the eye back wall, close to the original optic nerve, also regenerating after cutting (personal communication).

In addition to the retina and brain, salamanders regenerate the spinal cord, as shown in Figure 4. The cell source is ependymal cells that exhibit the properties of radial glia. In mature tissue, these cells express the key TFs that constitute the Dorsal/Ventral transcriptional factor code and morphogens and proliferate near the injury area. This causes the formation of a tubular structure, with the so-called terminal vesicle (whose cells exhibit a proliferative and migration activity which leads to tubular outgrowth) formed in the terminal part. As a result, the neuroepithelium characterized by directed growth and differentiation is formed, and its correct morphogenesis completes the process of spinal cord regeneration [95]. 

Low rates of somatic development, metabolism, and cell proliferation and differentiation form the basis of cellular natural “youth”. After analyzing the juvenile properties on different levels of the Urodela organization, as well as the broad regenerative capacities of these animals, a question arises as follows—can this knowledge be used to induce juvenility, i.e., rejuvenate cells of higher vertebrates in order to increase their regenerative activity? In regard to potential endogenous cell sources of regeneration in mammals, there are presumably two ways to make it feasible. The first is the use of already known and active extracellular and intracellular factors of Urodela that make up a regulatory network similar to the developmental one in these animals. The second is molecular/genetic and epigenetic modifications in accordance with the features of gene expression and epigenetic landscape characteristics of caudate amphibians. We can theoretically assume a combination of these two approaches above. However, for that we need to have more information about both possible gene losses in evolution [30], and the epigenome not yet studied in Urodela.

## 5. Attempts at Experimental Cell “Rejuvenation” to Stimulate Involvement of Mammalian Cells in Tissue Regeneration

We leave extensive information on obtaining induced stem cells from human fibroblasts and their use in regenerative therapy beyond the scope of the present review. This important field of research, being widely elucidated in the literature, does not, however, rule out other opportunities and approaches discussed here.

It is obvious that cell “rejuvenation” involves the removal of cells from signaling that stabilizes differentiation and their exposure to a new signaling that dictates dedifferentiation and re-entry in the proliferative phase. The term “retrodifferentiation” is proposed, which means the return of cells to the state of progenitors capable of proliferation and showing a low differentiation level [96]. Such cells, along with stem cells, can become a regenerative potential lost during the far-gone, terminal differentiation and cell aging in higher vertebrates.

Currently, the search for directed rejuvenation is carried out for certain cell populations and involves obtaining a younger cell phenotype through either gene modifications or modified extracellular and cell-to-cell signaling, as shown in Figure 5. For example, genes encoding growth differentiation factor 11 (GDF11, belonging to the activin-transforming growth factor β superfamily) were delivered to cells of aging mouse myocardium. The imposed overexpression of GDF11 was found to result in rejuvenation and increase the activity of myocardial stem cells (cardiomyocytes). Thus, the expression of aging markers p16 and p53 decreased, and cell proliferation increased. According to the work [97], the imposed expression of GDF11 in the mouse myocardium increases the cells’ capability of regeneration and wound healing after ischemia/reperfusion.

With approaches based on genetic modifications, it should be taken into account that the dysfunction of the genetic rejuvenation programs poses the risk of neoplastic alterations. Thus, the double knockdown of the tumor suppressor genes RB and ARF, which contributes to dedifferentiation and rejuvenation of muscle cells, should only be short transit to prevent the emergence of neoplasms caused by inhibition of oncogenesis suppressors [98]. It is worth mentioning that the newt genome lacks genes coding ARF proteins associated with tumor-cell invasion. In this regard, the newt muscle cells that enter the active proliferative phase during the limb and tail regeneration are assumed to be intrinsically more responsive to dedifferentiation signals than mammalian muscle cells [99].

Another approach that leads to cellular rejuvenation is creating the required conditions for the cell environment (in review [79]). In the work of Pei [100], incubation of chondrogenic cells under certain oxygen concentrations, a set of growth factors, and extracellular matrix (ECM) components, caused the proliferative activity and chondrogenic potential of chondroblasts to increase. It is emphasized that such an approach to cell proliferation stimulation and regeneration is less dangerous than genetic manipulation [100]. The environmental conditions that can restore the “young” properties to satellite muscle cells are currently being studied [11]. It has been found that the suppression of satellite cell activity with age can be eliminated via directed alterations in the Notch, MAPK, and TGF-β regulatory signaling pathways. To create a “young niche” for satellite cells, it is also suggested to use embryonic cells that produce “pro-regenerative” soluble signaling molecules activating the MAPK pathway [101].

The reviews [102,103] discuss research in which attempts were made at recovery of ageing myocardium through induction of cell “rejuvenation”. These reviews consider both genetic modification and effects of various viability and growth factors, such as follistatin-like 1 (FSTL1), growth-differentiation factor 11 (GDF11), and insulin-like growth factor 1 (IGF-I).

An example of efforts to achieve “retrodifferentiation” of CNS cells can be works based on heterochronic parabiosis. Ruckh and coauthors [104] demyelinated nerve fibers by increasing the activity of oligodendrocyte precursor cells exposed to young mouse monocytes used for myelin debris clearance. These studies indicate that enhanced remyelinating activity requires not only youthful monocytes, but also soluble serum factors [104].

The systemic environment has a direct effect on the neurogenesis at the expense of neural stem cells in the adult mammalian brain [105]. Neurogenic niches are assumed [106] to have factors required for both proliferation of progenitor cells and their specialization and integration into the system of pre-existing neurons. It was shown that under conditions of heterochronic parabiosis, the blood of old mice caused the proliferation in neurogenesis zones to decrease, while the blood of young animals, on the contrary, reduced the number of differentiating neurons [107,108,109]. An attempt was made to induce “retrodifferentiation” of Müller glia cells, which are a potential regeneration source in the mammalian retina. Peng and coauthors [110] used mouse embryonic stem cells as a source of regulatory factors. Human retina Müller cells were incubated in vitro with mouse embryonic stem cell-derived extracellular vesicles (mESEVs). The effect of mESEVs that were in a conditioned medium was manifested as the induction of dedifferentiation and reprogramming of Müller cells, which resulted in the appearance of neurogenesis signs [110]. Previously, it was shown that exposure of cultured human Müller cells to mESEVs induces transcriptome changes in these cells, i.e., the reactivation of the program involved in retinogenesis during development [111].

In their study, Peng and coauthors [110] separated the microvesicle and exosome fractions, determined their contents, and showed differences. A single exposure to mESEVs changed the levels of OCT4 and SOX2 in cultured Müller cells, while repeated treatment reduced the expression of vimentin (the marker specific protein of the glial cell cytoskeleton), but increased the expression of early retinal transcription factor Pax6. Variations in the expression level of OCT4, SOX2, and PAX6 mRNAs occurred when cells were exposed to a medium that contained vesicles but not exosomes. However, the necessity of in vivo experiments is evident, as we need to understand the interaction of the contents of extracellular vesicles with the native microenvironment of Müller cells in situ.

The use of cell-derived vesicles, a heterogeneous population of particles that may include exosomes, microvesicles, ectosomes, membrane particles, exosome-like vesicles, and apoptotic vesicles, to regulate cell differentiation plasticity, increase cell viability, wound healing, regeneration, and antitumor therapy is currently an actively developed field of research (reviews [112,113]). Much attention is paid to exosomes, which are 40–100 nm intraluminal vesicles containing proteins, RNAs, microRNAs, and lipids and are capable of stimulating cell rejuvenation and cell regeneration potential. The source of exosomes is often stem cells. The action of exosomes substitutes the well-known bioactivity of stem cells, and being, in fact, an encapsulated set of biological ingredients, provides promising prospects [114].

Thus, the major idea underlying many approaches that are currently being developed to enhance regenerative potencies is, in particular, the experimental creation of an exogenous “pro-youthful” environment and giving the cells a young molecular/genetic status corresponding to this environment. Despite maturation, stabilization of differentiation, and aging, cells are still capable of responding to the signals produced by the new environment. The source of regulatory factors, besides embryonic and stem cells, may probably be cells and tissues of pedomorphic Urodela, some terrestrial, undergoing direct development species of salamanders [49,115] and other Anamnia with high regenerative activity. This assumption is supported by existing views that the regenerative potential is present in the genome of not only lower vertebrates, but also in mammals and humans, and despite the mechanisms for its implementation being blocked, they are still retained [116]. There is also a suggestion that the enhancers of the work of genes responsible for the processes that constitute regeneration are evolutionarily conserved among species [31].

## 6. Tissues of Animals with High Regenerative Capacities as a Source of Compounds for Stimulating Regeneration in Higher Vertebrates

As it is commonly known, the regenerative activity of adult mammalian and human cells is low. In humans, partial regeneration is possible in response to damage in bones, muscles, peripheral nerves, epithelial tissues, blood, i.e., in tissues where aging or dying cells are renewed and replaced by newly differentiating stem cells, which are also subject to aging [117].

Regeneration of tissues and organs in caudate amphibians occurs due to stem cells, as well as specialized cells that exhibit a high degree of differentiation plasticity, which can be explained, in particular, by the retention of a number of above-considered juvenile traits by animals. This group of animals and also some fish species that possess the ability to regenerate certain tissues can become producers of factors inducing the destabilization and decrease in the differentiation level (rejuvenation) of cells that are a potential source for tissue regeneration in higher vertebrates.

A few examples of the approach described above exist. One approach attempted to initiate in vitro dedifferentiation of C2C12 mouse myotubes, having muscle differentiation, exposed to raw extracts from newt regenerating limb tissues [118]. As a result, the initial differentiation was inhibited—the level of expression of the muscle marker proteins MyoD, myogenin, and troponin T was significantly reduced. In this case, muscle tubules can produce small forms of myotubes and proliferating mononuclear cells, a source of muscle regeneration in vivo. This work has demonstrated the in vitro dedifferentiation of mammalian cells with the formation of blast cells in the same way as it occurs in the case of newt limb blastema formation in vivo. Earlier, blood serum, thrombin and some of its fractions, or a mixture of thrombin and serum were used for this purpose [119]. However, these substances might cause myonuclei of tubules to enter the cell cycle, but no proliferation with the production of certain mononuclear cells occurred. According to the observation of McGann and coauthors [118], the factors that worked in their study had a protein nature, were in the soluble fraction of the extract, heat-resistant, and trypsin-inhibited. The authors [118] suggested these factors in the extract from regenerating newt limb to be extracellular proteins that acted as ligands for receptors transmitting signals to cell dedifferentiation in vivo.

There are examples of reductions in the level of mammalian cell differentiation using protein extracts from fish tissues. Thus, in the study of Kim et al. [120], the use of blastema extract from the regenerating tail of the teleost fish *Sternopygus macrurus* also had an effect on C2C12 myoblasts in vitro, but somewhat different from that described in the work cited above [118]. The treatment with the extract did not induce the entry of myonuclei in the cell cycle but led to the isolation of myoblasts and inhibition of their differentiation. Subsequently, the fusion de novo of isolated myoblasts into myotubes was observed [120]. The above studies indicate the probability of regenerative responses of mammalian cells exposed to substances extracted from regenerating tissues of lower vertebrates such as newts and fish. Several publications show that not only the methods of “release” and then “rejuvenation” of cells may differ, but also the patterns of effects of regenerative extracts on primary muscle cells and cell lines [121]. The extract obtained from regenerating newt limb after repeated amputation has a greater effect on dedifferentiation and proliferation of cultured C2C12 cells compared to that of the extract after the first amputation [121]. However, there is evidence of a toxic effect of regeneration (blastema) extracts from newt limbs and fish tails on C2C12 mouse cells [122].

In their study, Chen and coauthors [123] have found that the extracellular matrix (ECM) of zebrafish heart tissue can induce heart regeneration in mammals. The heart regeneration in adult zebrafish is known to occur even when 20% of tissue is removed through ventricular amputation, while mammals do not show this ability [124]. In mice, this ability is lost with age—neonatal animals are still able to restore 10% of the heart apex after resection, whereas a scar is formed at the site of damage in adult mice [123]. ECM was obtained from intact fish and fish on day 3 after amputation of a heart ventricle fragment via decellularization of this tissue. The effectiveness of ECM was evaluated both in vitro, towards cultured human cardiac precursor cell populations, and in vivo, when it was injected into mice after an experimentally induced heart failure. A single injection of heart tissue ECM obtained from zebrafish to mice not only induced heart regeneration after a heart failure, but also restored its function. The study revealed the proliferative activity of cardiomyocytes, an increase in the dedifferentiated progenitor cells population, and reactivation of ErbB2 (epidermal growth factor receptor-2, receptor tyrosine kinase) expression. Under in vitro conditions, fish ECM had pro-proliferative and chemotactic effects on human cardiac precursor cell populations [125]. In general, these results show that cell-free ECM obtained from animals which are much lower on the evolutionary scale and have a higher regenerative capacity can activate the regeneration of mammal tissue that is not capable of regeneration.

As we mentioned above, vesicles are present in the extracellular space [126]. They contain microvesicles that detach off from the cell surface and are surrounded by a double membrane, and also exosomes of the endosomal origin. Extracellular vesicles can transmit the contents (mRNA, microRNAs, functional proteins, and lipids) to neighboring cells, being a component of intercellular communication [126]. Extracellular vesicles have recently been visualized in a regenerating tissue of the zebrafish caudal fin. The main goal of the study [127] was to demonstrate the involvement of extracellular vesicles in intercellular communication during regeneration. The authors used an in vivo electroporation method that does not disturb the regeneration process. Transferred plasmids bearing exosomal markers showed mosaic expression in the regenerating caudal fin blastema, which indicated the interaction of blastemal cells with other cells via vesicles during regeneration [127].

One recent work, for the first time, studied extracellular vesicles obtained from a conditioned medium after culturing blastemal cells of the newt regenerating limb tissues in it [128]. These structures resembled mammalian exosomes in many features, such as size (~100–150 nm), content, surface antigens, etc. This indicated the similarity of the mechanisms of biogenesis and extracellular vesicles secretion in vertebrates from different classes. The incubation of mammalian cardiomyocytes with extracellular vesicles, released into the medium by blastemal cells of newt regenerating limb tissues, caused changes in gene expression. This, in turn, increased the cardiomyocytes’ resistance to oxidative stress and cell viability through an increase in the PI3K/AKT signaling pathway, the key proteins of which are PI3K (phosphatidylinositol 3-kinase) and Akt (Protein Kinase B). A noteworthy fact is that mRNA transcription factors (HOXC6, FOXQ1, and SOX1), the regulators of early development and nervous system histogenesis, were found in the contents of newt extracellular vesicles; the latter proved to be generally richer in RNA and proteins compared to mammalian exosomes. A presence of a large number of mRNAs that encode nuclear receptors and membrane ligands was detected [128]. This gave reason to assume that the factors in the extracellular vesicles of the newt blastemal vesicles may be efficient in altering properties of mammalian cells.

According to preliminary data [129], a raw extract from the newt retina can induce dedifferentiation and proliferation of albino rat (2-month-old) RPE cells, a potential source of retinal regeneration in mammals [54], under conditions of organotypic cultivation in vitro. Proliferation in rat RPE was detected using the PCNA marker protein and through estimating the mitotic index. As known, rat RPE cells, both intact and under in vitro conditions, exhibit low proliferative activity. However, in the case of exposure to newt retinal extract, the number of mitotic and PCNA-labeled cells increased significantly. Rat RPE cells, when dividing, proved to be capable of producing an additional outer row of cells, as it occurs during the retinal regeneration from newt RPE in vivo and in vitro. Commercial growth factors EGF and FGF2 had a similar, though to a lesser extent, effect on changes in the phenotype and proliferative activity of albino rat RPE cells cultured in organotypic culturing. Furthermore, FGF2, which is known for its mitogenic effect during regeneration and eye development, was earlier found in the adult newt retina [130]. RPE of 2-month-old albino rats (that responded with dedifferentiation and proliferation to treatment with newt retinal extract) showed signs of aging [131].

Recently, we set up experiments on the ARPE-19 line of the human eye RPE cells in vitro exposed to media conditioned by the newt regenerating retina. The results of the study [132] provided interesting information that, nevertheless, requires further study and analysis. During 120 h of cultivation with exposure to a medium conditioned by the newt retina, the shape of ARPE-19 cells changed, and their size increased. These changes were accompanied by a decrease in the proliferative activity evaluated by the MTT test (the Cell Proliferation Kit for nonradioactive quantification of cellular proliferation). At an early stage of cultivation (24–48 h), quantitative PCR showed a decrease in expression of mRNA genes such as *OCT4, NANOG, OTX2, BMP2, BMP4*, and *GSK3B*, followed (at 48–72 h) by a decrease in mRNA of transcription factor KLF4. Simultaneously, there was an increase in the expression of mRNA genes of the pro-neural marker protein TUBB3, and also an increase in the level of mRNA CCND1, a specific cell cycle regulator [132]. Thus, a short-term exposure to the medium conditioned by regenerating newt retina extract destabilized, but without eliminating, the epithelial differentiation of ARPE-19 cells, and also induced the manifestation of traits characteristic of the early neural differentiation stages. These data allow careful drawing of an analogy between ARPE-19 cells, in vitro exposed to media conditioned by the regenerating newt retina, and native newt RPE cells, which, as we mentioned above, exhibit a gene expression combining the characteristics of both specialized cells and early progenitor cells. The information presented in this section, as shown in Figure 6, is the primary data that require further research in two main directions: (1) to study the pattern of effects of the factors produced by regenerating tissues of lower vertebrates on mammalian and human cells; (2) to study the range of these factors and identify among them the key regulators of rejuvenation through comparing them with the already known exogenous factors that can stimulate tissue regeneration in vertebrates.

## 7. Conclusions

Caudate amphibians are animals that exhibit the highest regenerative abilities among vertebrates. The heterochrony and pedomorphosis that are characteristic of their development have resulted in a number of unique features manifested on all levels of organization, from organismal to molecular. The evolutionary history of these animals has formed a very large-sized genome, with a minimum of mutations. The size of the genome determined the large cell sizes, the low rate of proliferation, and a decrease in the rate/cessation of tissue development with the onset of premature puberty. This led to a simplification of the structure and a decrease in the level of differentiation of certain tissues such as, in particular, the brain, the spinal cord, and the retina. Regions of constant cell growth and populations of cells with a high degree of plasticity and regenerative activity have been recorded from eyes and brain, and also from some other tissues [133,134]. The expression of genes encoding TFs, which are regulators of early development, is detected on the molecular and genetic level in tissues, including the brain and retina, of mature Urodela. An assumption has been made that the high tissue regeneration ability results, in particular, from the natural longlife juvenility of Urodela.

The factors responsible for the decrease in regenerative abilities with age and in the evolutionary series are still poorly understood. They are presumably associated with (1) a molecular genetic signature; (2) epigenetic modifications that prohibit the expression of genes responsible for dedifferentiation and reprogramming; (3) the cell environment stabilizing the differentiation, which is nonpermissive for regenerative responses. All of these factors depend on aging in terms of specialization depth and phenotype plasticity inhibition, and also on cell aging proper. Thus, the idea underlying the approach being discussed in the review is to create a pro-youthful exogenous environment to give cells a “young” molecular genetic status. There is increasing evidence for the ability of terminally differentiated cells, as well as aging stem cells, to respond to “pro-youthful” signals generated by the microenvironment. Epigenetic mechanisms also play an important role in these responses in parallel.

The identified relationship of the regenerative abilities with the phenomenon of natural longlife juvenility in Urodela may help resolve many issues. These include obtaining information about the factors necessary to maintain the plasticity properties in specialized cells and populations of low differentiated progenitor cells of adult animals, and also the factors regulating their differentiation. Study of the latter factors, which can be produced by both intact and regenerating tissues of caudate amphibians and some fish species, provides a promising approach to “rejuvenation” of higher vertebrates’ cells to be involved in the regeneration processes. The experiments conducted on cell and tissue models in vitro and in vivo brought the first interesting but not unambiguous results. It should be noted that the factors included in the “management” of regeneration processes in amphibians are strictly regulated to prevent uncontrolled plasticity that could destabilize tissues and/or lead to cancer or pathologic cell plasticity [135]. The relationships with the own signaling environment of mammalian cells, in which the factors produced by tissues of evolutionarily distant species would enter in vivo, and whether they exhibit the specificity and desired effect, remain unknown. Hopefully, answering these questions will open up new opportunities for stimulation of regeneration in mammals and humans.

## Figures and Tables

**Figure 1 jdb-09-00002-f001:**
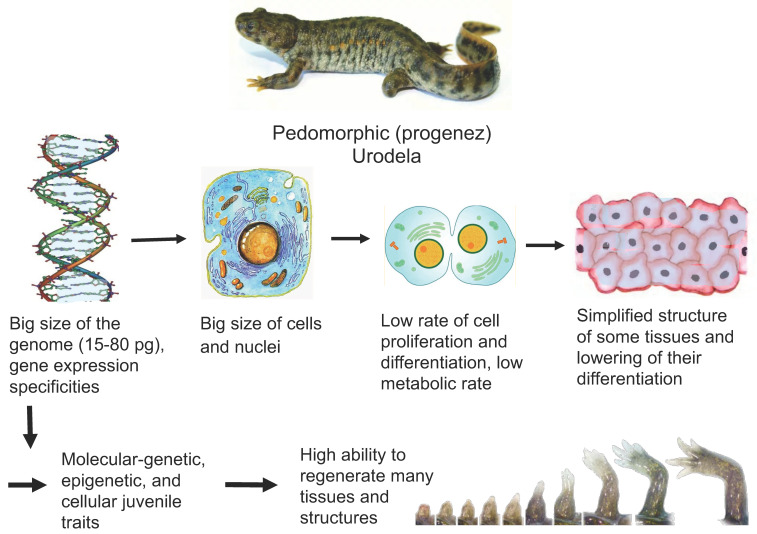
Features of Urodela associated with their high regenerative ability. Description presents in the text.

**Figure 2 jdb-09-00002-f002:**
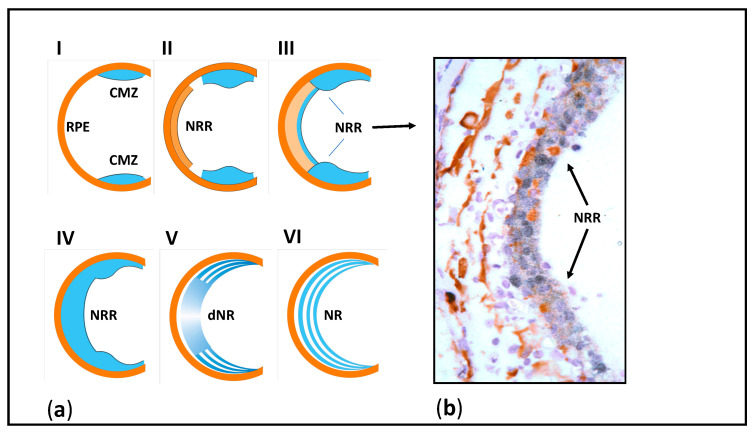
(**a**) Stages of retina regeneration by retinal pigment epithelium cells after surgical removal of the retina in the newt. RPE—retinal pigment epithelium, CMZ—circumferential marginal zone of the retina, NRR—neural retinal rudiment (blastema), dNR—differentiating neural retina, NR—newly formed neural retina. (**b**) Histological picture of retinal regenerate (arrows). Retinal rudiment cells have a high synthetic activity (black nuclei—intense inclusion of ^3^H-labelled tryptophan).

**Figure 3 jdb-09-00002-f003:**
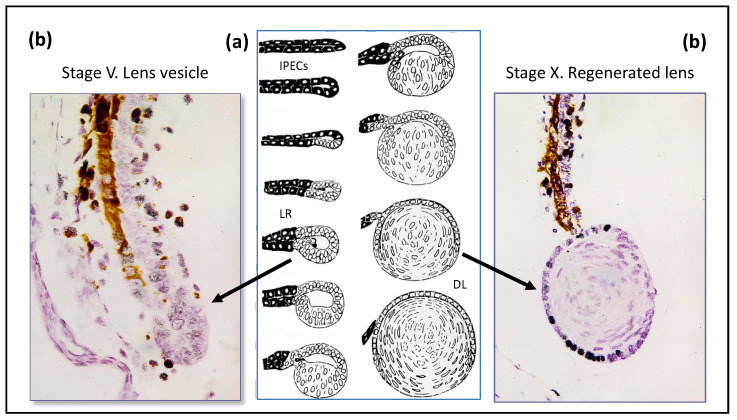
(**a**) Stages of regeneration of eye lens from the dorsal part of the iris after lens surgical extirpation in the newt. IPECs—iris pigment epithelial cells, LR—lens regenerate, DL—differentiated lens. (**b**) Histological pictures of newly forming lens (grey and black nuclei—inclusion of ^3^H- thymidine labelled cells).

**Figure 4 jdb-09-00002-f004:**
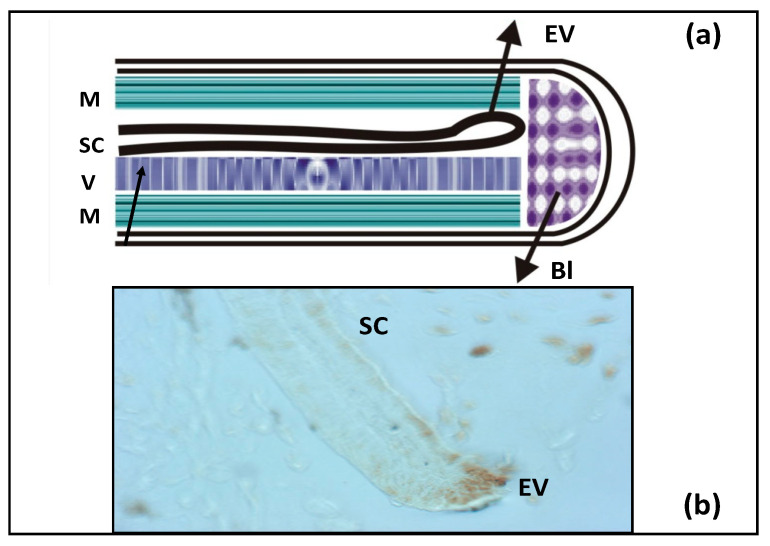
(**a**) Schematic diagram of tail regeneration in the newt. SC—spinal cord, EV—ependymal vesicle, V—vertebra, M—muscles, Bl—blastema. (**b**) Histological picture of growing SC with EV on the edge. EV cells proliferate (brown color—BrdU-positive cells).

**Figure 5 jdb-09-00002-f005:**
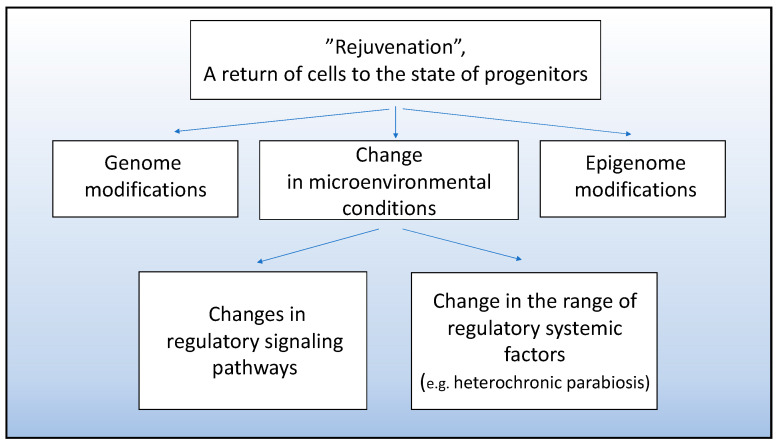
Approaches to obtaining cell populations with young phenotype and progenitor properties.

**Figure 6 jdb-09-00002-f006:**
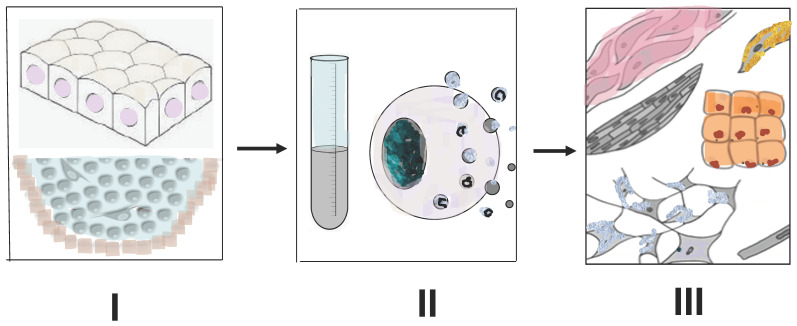
Main steps of an approach to stimulate tissue regeneration in higher vertebrates by the use of extracellular material of animals which have high regenerative potential. **I**—using of intact or blastemal tissues of amphibia and fish capable of regeneration; **II**—obtaining of cell and tissue extracts, secretomes, extracellular vesicles, exosomes; **III**—study of their effect on target cells and tissues of higher vertebrates, incapable of regeneration. Description presents in the text.

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
