# Peer review of "Study of Natural Longlife Juvenility and Tissue Regeneration in Caudate Amphibians and Potential Application of Resulting Data in Biomedicine"

_jdb, 2021, doi:10.3390/jdb9010002_

Round 1
Reviewer 1 Report
Review of JDB 1049629
There is a lot of discussion about regeneration and great interest in trying to get other organisms, especially humans, to be able to approximate what seems to come easily to caudate amphibians -- the salamanders. Here the author suggests that heterochrony and pedomorphosis is key. But that begs the questions: what is pedomorphosis? what is its biological basis? The author focuses on juvenality and attributes it, at least in part, to large genomes and large cells and consequences. The approach is largely phenomenological, but there are interesting foray into possible mechanisms. I find this all very interesting and stimulating and commend the author for the analytical review. A biological explanation for the phenomena discussed here has recently been proposed in a paper accepted for publication in Developmental Dynamics (by Sessions and Wake), and it will appear on the journals website within days. I recommend that the author consult that paper.
In general this is an excellent review and complementary in many ways to the new paper cited. This review considers a broader range of work and is a very useful overview of what is understood concerning salamander regeneration.
I think the conclusion is appropriate and stimulating.
Because this is a review I choose not to nit-pick the examples cited, which sometimes have some alternative explanations not explored here.
In the lines below I offer some specific observations and suggestions.
Line 26 This is a controversial statement: THE KEY ISSUE? I doubt it. A key issue? Perhaps.
Line 36 Puberty is not a term used by amphibian biologists.
l 42 This is where the Developmental Dynamics paper becomes especially relevant.
l 62 "higher" and "lower" are archaic scientific terms and are no longer used in phylogenetics.
l 176 cells
l 181 This section is especially relevant to the Developmental Dynamics paper
l 195 For a general framework for heterochrony see Alberch et al. 1979 Paleobiology.
l 200 A new paper reports a low value of 9.3 in a minute salamander, Thorius, from Mexico (Decena-Segarra et al. 2020 American Naturalist).
l 257 A long list of papers from the Roth lab is relevant.
l 290 I recommend deleting useless words such as "it should be noted that--"
l 304 et seq. Surely large genome size, leading to large cell, and subsequent low rate of cell division is the ultimate cause
l 370 et seq. The main point seems to be obvious but not stated: salamander cells have been through many fewer rounds of cell division than those of other vertebrates. And each cell has lived far more slowly. They are naturally "young", except chronologically.
l 378-381. I do not find this useful or explanatory.
l 431 Not "a research". Just "research"
l 437 Not "could perform" Just write "Ruckh et al. [107] demyelinated nerve fibers by --"
l 442 I do not understand "systemic environment"
l 459 Not "could separate". Just "separated"
l 464 Not "it was found that". Just "Variations in the expression level--- occurred"
l 479 et seq. An important summarizing conclusion. Could be highlighted more. I agree with the author that this is a key finding. But it is not just "pedomorphic" salamanders -- remember the important work of Sessions and Larson (1987), which found that direct developing tropical salamanders regenerate as well as any. For a modern treatment see Arenas Gómez et al., 2017, Regeneration DOI: 10.1002/reg2.93
l 508 Write "A few examples of the approach described above exist. One approach attempted to initiate --".
l 536 Just "Several publications show that--"
l 538 Delete useless words: " It is also worth mentioning that"
l 554 Delete " It was found that"
l 592 Awkwardly written. Try "--observed [130], suggesting that the factors--"
l 604 Delete "It is worth mentioning that"
l 609 Delete "It is important to note that"
Author Response
Reviewer 1. I express my sincere appreciation for the clarifications and comments made, as well as for additional useful and interesting information and references to the most recent works.
Below there are changes I made to the text . Links to them are placed in the margins of the manuscript.
Line 26. This is a controversial statement: THE KEY ISSUE? I doubt it. A key issue? Perhaps. - Sure, there are some other “key issues” in biology and medicine, so article “a” is more appropriate.
Line 36. Puberty is not a term used by amphibian biologists. - I did try to find more correct term. What I could find, it was “mature”. So “puberty” was replaced in the text by ”being mature”.
l 42 This is where the Developmental Dynamics paper becomes especially relevant. - As of today, unfortunately, I haven't found a recommended paper by Sessions and Wake in Developmental Dynamics (I looked through last issues on the site of the Journal). The paper definitely will be taken into account in future discussions on these issues.
l 62. "higher" and "lower" are archaic scientific terms and are no longer used in phylogenetics. - Indeed, the terms “higher” and “lower” are quite archaic but these definitions are still common to a certain range of biologists. I deleted “low and higher vertebrates” in the abstract and somewhere in the main text.
l 176. cells - corrected
l 181. This section is especially relevant to the Developmental Dynamics paper. - As of today, unfortunately, I haven't found a recommended paper by Sessions and Wake in Developmental Dynamics (I looked through last issues). The paper definitely will be taken into account in future discussions on these issues
l 195. For a general framework for heterochrony see Alberch et al. 1979 Paleobiology. - I could not include the information and link to the paper (Alberch et al., 1979) in the text because I did not find it in the databases available to me, perhaps the text of the article is not digitized.
l 200. A new paper reports a low value of 9.3 in a minute salamander, Thorius, from Mexico (Decena-Segarra et al. 2020 American Naturalist). - It is interesting, thank you! Since the opportunities to sequence salamander genomes appeared recently, I am sure many more refinements and discoveries will be made along the way. However, the previously made assessment of them as ones of the largest seems to remain.
l 257. A long list of papers from the Roth lab is relevant. Thank you for your helpful recommendation. It is difficult to take into account the array of previously published works for each of the subsections of the review. They will be considered in a more in-depth analysis of the specific issues that have just been raised.
l 290. I recommend deleting useless words such as "it should be noted that--" Useless words were deleted through the text.
l 304 et seq. Surely large genome size, leading to large cell, and subsequent low rate of cell division is the ultimate cause - Being agree with this statement I added the next sentence at the end of chapter. “However, in any case, the ultimate cause seems to be the large size of the genome leading to a large cell size and subsequent low rate of cell division”.
l 370 et seq. The main point seems to be obvious but not stated: salamander cells have been through many fewer rounds of cell division than those of other vertebrates. And each cell has lived far more slowly. They are naturally "young", except chronologically. – “Low rate of somatic development, metabolism, cell proliferation and differentiation form the basis of cellular natural “youth”. – added to the chapter
l 378-381. I do not find this useful or explanatory. - Previously (Biomedicines, 2020) I compared the features of gene expression and the epigenetic landscape of cells-sources of regeneration of eye tissues in amphibia and mammals. In that paper it was suggested that these features are implemented to initiate and successfully regenerate the retina after damage in amphibians, but do not work in mammalian the same but potential cell sources. This was the basis for the assumption I made.
l 431. Not "a research". Just "research" – article “a” - removed
l 437 Not "could perform" Just write "Ruckh et al. [107] demyelinated nerve fibers by --" “could perform” – removed.
l 442. I do not understand "systemic environment". - I used this term to show that along with the effect of local micro-environment, there is also an influence from factors that work at the level of the whole animal body (serum factors, hormones, etc.). This definition I met in the literature and found it convenient. The sense of the term is explained above (lines 173 – 177 in the final text).
.
l 459. Not "could separate". Just "separated" – rewritten
l 464. Not "it was found that". Just "Variations in the expression level--- occurred" “It was found that” – deleted
l 479 et seq. An important summarizing conclusion. Could be highlighted more. I agree with the author that this is a key finding. But it is not just "pedomorphic" salamanders -- remember the important work of Sessions and Larson (1987), which found that direct developing tropical salamanders regenerate as well as any. For a modern treatment see Arenas Gómez et al., 2017, Regeneration DOI: 10.1002/reg2.93 - A clarification was made, additional references (Sessions, Larson, 1987; Arenas Gomez et al., 2017) were added.
l 508. Write "A few examples of the approach described above exist. One approach attempted to initiate --". Sentences were rewritten.
l 536 Just "Several publications show that--" – the sentence was rewritten
l 538 Delete useless words: " It is also worth mentioning that" -deleted
l 554 Delete " It was found that"- deleted
l 592 Awkwardly written. Try "--observed [130], suggesting that the factors--" – Sentence reformulated
l 604 Delete "It is worth mentioning that" - deleted
l 609 Delete "It is important to note that" - deleted

Reviewer 2 Report
REVIEW
Study of natural longlife juvenility and tissue regeneration in caudate amphibians and potential application of resulting data in biomedicine
This work is a review of the regenerative capacities in urodele amphibians and the molecular basis of this capacity are discussed
Urodeles have the highest regenerative abilities among vertebrates due to their peculiar tissue arrangement, funded in specific molecular , cellular and genomic characteristics most probably caused by the neotenic properties of these animals.
All of them led to a low level of cell differentiation and a high degree of plasticity and regenerative activity in most urodele tissues as eyes, limbs, brain and spinal cord. One of the most interesting features is the constant expression of some developmental TFs such as Pax or Sox that likely provide them a continued activated gene regulatory network for a natural longlife juvenility. In the other hand, the known aging factors mostly converge in cell specialization and plasticity inhibition (molecular genetic signature, epigenetic modifications the cell environment stabilizing the differentiation). They are vaguely understood but seems that they would be reverted by “pro-youthful” signals
This review identify the relationship of the factors necessary to maintain the plasticity the regenerative abilities for maintaining the young status in proliferative and differentiated cells, factors that regulate the regeneration and dedifferentiation and recopilate the information about factors, which can be produced by both intact and regenerating tissues of some urodeles and fishes species involved in the regeneration and rejuvenation. They also discuss the molecular implications and side effects of these rejuvenating molecules in higher vertebrates tissues
COMMENTS TO THE AUTHORS:
This is a wonderful and interesting review that is based on reliable previous works, many of them signed by the author. So, it is a good advantage having a recent review in this quickly developing area.
I have only good sensations reading it and my best words for the author
I have only some esthetic observations that I have included as comments in the attached pdf

Author Response
Reviewer 2
First and foremost, I express my great appreciation to the reviewer (2) for the careful reading and suggestions to make adjustments to the text and drawings. Below there is a list of the corrections I made. References to them are placed in the margins of the manuscript.
Line 27 – changed to “a key”
34-35 – reformulated “Furthermore, there are tetrapods which being mature are able to regenerate tissues and this ability is not lost with age”.
45 – all necessary references are provided below, in the chapters corresponding to each separate points. Otherwise, reference list in the paragraph will be extremely long and hard for finding necessary particular information.
235-236 – Figure 2. Replaced by new one, more “readable” and in color.
311-312 – Figure 3. Replaced by new one, more “readable”.
365-366 – Figure 4. Changed for better reading of morphology.
383-384 – New figure (number 5) and its legend are added (as recommended).
List of references has been checked, text type and color unified, latin names of animal species are in italics now.
